# Genome-Wide Identification of the Soybean LysM-RLK Family Genes and Its Nitrogen Response

**DOI:** 10.3390/ijms241713621

**Published:** 2023-09-03

**Authors:** Kaijie Yao, Yongliang Wang, Xia Li, Hongtao Ji

**Affiliations:** 1National Key Laboratory of Crop Genetic Improvement, College of Plant Science and Technology, Huazhong Agricultural University, Wuhan 430070, China; ykj@webmail.hzau.edu.cn (K.Y.); wangyonglaing@webmail.hzau.edu.cn (Y.W.); xli@mail.hzau.edu.cn (X.L.); 2Hubei Hongshan Laboratory, Huazhong Agricultural University, Wuhan 430070, China

**Keywords:** soybean, LysM-RLK, rhizobia, nitrogen, gene expression

## Abstract

Lysin-Motif receptor-like kinase (LysM-RLK) proteins are widely distributed in plants and serve a critical role in defending against pathogens and establishing symbiotic relationships. However, there is a lack of comprehensive identification and analysis of LysM-RLK family members in the soybean genome. In this study, we discovered and named 27 *LysM-RLK* genes in soybean. The majority of LysM-RLKs were highly conserved in Arabidopsis and soybean, while certain members of subclades III, VI, and VII are unique to soybean. The promoters of these *LysM-RLK*s contain specific *cis*-elements associated with plant development and responses to environmental factors. Notably, all *LysM-RLK* gene promoters feature nodule specificity elements, while 51.86% of them also possess NBS sites (NIN/NLP binding site). The expression profiles revealed that genes from subclade V in soybean roots were regulated by both rhizobia and nitrogen treatment. The expression levels of subclade V genes were then validated by real-time quantitative PCR, and it was observed that the level of *GmLYK4a* and *GmLYK4c* in roots was inhibited by rhizobia but induced via varying concentrations of nitrate. Consequently, our findings provide a comprehensive understanding of the soybean *LysM-RLK* gene family and emphasize the role of subclade V in coupling soybean symbiotic nitrogen fixation and nitrogen response.

## 1. Introduction

Plants reside in a microbial-rich environment and must interact with a large number of pathogens and symbiotic bacteria [1]. The initiation of the defense response of plants to pathogen infection and the establishment of symbiotic relationships between plants and symbiotic bacteria depend on the molecular communication between interacting organisms [2]. Within the case of pathogen interactions, plants recognize molecular-associated molecular patterns (MAMPs) through pattern recognition receptors (PRRs) on their cell membrane, such as receptor-like proteins (RLPs) and receptor-like protein kinases (RLKs), to activate endogenous immunity [3,4]. RLKs contain the intracellular kinase domain, transmembrane domain, and extracellular ligand binding domain, which are involved in signal transduction, while RLPs lack intracellular regions. Similarly, when interacting with symbiotic bacteria, plants can recognize signal molecules synthesized by symbiotic bacteria via specific RLKs or RLPs receptors to establish symbiotic relationships [4,5,6]. Plant RLKs include many types, among which LysM-RLK plays an important role in defense response initiation and root endosymbiosis establishment [3,4,6]. The LysM motif, spanning approximately 40 amino acids in length, exhibits a conserved βααβ three-dimensional structure, facilitating its binding to chitin, peptidoglycan, lipopolysaccharide, and other MAMPs, thereby triggering an immune response or symbiosis initiation. Nodulation/Nod factors (NFs) and Mycorrhization/Myc factors (MFs) are chitin-related oligosaccharides, which can be recognized by LysM-RLK on plant cell membranes to initiate symbiotic signals [2,7,8]. The kinase domain of the LysM-RLK protein usually contains some conserved motifs or residues, including the G-rich motif (phosphate-binding glycine), AxK motif (phosphotransfer), HRD motif (catalytic loop), DFG motif (Mg-binding loop), and PE motif (activation zone), which are all necessary for ATP hydrolysis [9,10]. According to the characteristics of the intracellular kinase domain of LysM-RLK, Lys-RLKs can be categorized as LysM-RLK-I/LYKs and LysM-RLK-II/LYRs. LysM-RLK-I/LYKs feature a canonical kinase domain capable of autophosphorylation in vitro, while the LysM-RLK-II/LYRs comprise a pseudo-kinase domain lacking one or more active kinase motifs [2,11].

Genetic evidence demonstrates that chitin perception is considered necessary in the process of pathogen infection to properly activate the defenses that limit pathogen invasion. In *Arabidopsis thaliana* (*A. thaliana*), five Arabidopsis LysM-RLKs have been identified so far [9,10]. AtLYK1 (also known as AtCERK1) acts as a chitin receptor, essential for chitin-induced immune responses, while AtLYK5 serves as a co-receptor with AtLYK1, transmitting extracellular signals to the intracellular, properly activating the defenses. AtLYK4, which is evolutionarily related to AtLYK5, also functions in chitin immune signal, demonstrating functional redundancy with AtLYK5. AtLYK2 is essential for chitin sensing and early signaling events, and AtLYK3 acts as a negative regulator of the basal expression of defense genes, although the specific MAMPs or signal cues for AtLYK2 and AtLYK3 remain unidentified [12,13,14,15]. In *Oryza sativa* (*O. sativa*), *OsCERK1*, the orthologous of *AtCERK1*, encodes a LysM-RLK with a canonical intracellular kinase domain and is essential for chitin signal transduction. Although OsCERK1 does not directly bind to chitin, it interacts with OsCEBiP, which directly binds to chitin but lacks an intracellular domain, to transmit extracellular signals into the cell [16].

Apart from participating in plant immune responses, the LysM-RLK is also involved in the symbiotic establishment, including root nodule symbiosis (RNS) and arbuscular mycorrhizal symbiosis (AMS) [2]. In 2003, it was reported that LjNFR1/LjNFR5 and MtLYK3/MtNFP act as the Nod factor receptors of the model legume *Lotus japonicus* (*L. japonicus*) and *Medicago truncatula* (*M. truncatula*), respectively, and the extracellular LysM domain of LjNFR1 and LjNFR5 can directly bind to the Nod Factors (lipochitooligosaccharide). Furthermore, the N-terminal LysM1 domain determines the specificity of receptor–ligand binding [17,18,19,20,21,22]. Soybean Nod factor receptor genes *GmNFR1α*/*GmNFR1β* (orthologous to *LjNFR1* and *MtLYK3*) encode LysM-RLKs with canonical kinases, while other Nod factor receptor genes *GmNFR5α*/*GmNFR5β* (orthologous to *LjNFR5* and *MtNFP*) encode LysM-RLK with pseudo-kinases [23,24]. In contrast, *M. truncatula* LYK9 and *O. sativa* CERK1, as members of the LysM-RLK family, have been shown to function in regulating AMS [25,26].

Soybean, a crucial legume crop, satisfies approximately 35% to 50% of the global population’s protein requirements via RNS under soil nitrogen deficiency status. Upon detecting low nitrogen signals, soybean triggers a series of molecular and metabolic responses. It releases flavonoids into the rhizosphere to attract rhizobia, where the rhizobia then secrete NFs, which will be recognized by GmNFR1α/β and GmNFR5α/β to initiate the NFs signaling pathway, while a high concentration of nitrogen inhibits RNS establishment [17,18,27,28,29]. Given soybean’s importance and nitrogen’s significant effect on RNS, we utilized bioinformatics methods to scrutinize the role of LysM-RLK family members in the soybean genome. Our aim was to investigate the involvement of soybean Nod factor receptor genes and their homologs in symbiotic nitrogen fixation and nitrogen response, providing a theoretical foundation for cultivating novel varieties with an improved nitrogen fixation efficiency.

## 2. Results

### 2.1. Identification of Soybean LysM-RLKs

Soybean LysM-RLK receptor GmNFR1α/β and GmNRF5α/β were identified for the reorganization of rhizobia and the establishment of SNF [17,18]. However, the other soybean LysM-RLK family members are not identified and characterized in silico. To identify all LysM-RLK proteins in soybean, we searched the Pfam database using the LysM domain (PF01476) and the Kinase domain (PF00069) as query criteria. A total of 87 LysM domains containing proteins and 4454 Kinase domains containing proteins were obtained via a hidden Markov model (HMM) search. Among them, 48 proteins encoded by 25 genes contain both the LysM domain and the Kinase domain, and these 25 genes were initially identified as *LysM-RLK* genes (Appendix A). Furthermore, the known *A. thaliana* LysM-RLK protein sequences, which include At-LYK1/2/3/4/5 [9,10], were used as queries for a Blast search in the soybean database. All retrieved sequences with low reliable similarities (identity cut-off of 40% and e-value cut-off of 0.001) were removed. The remaining protein sequences were further verified via the NCBI CD-Search tool (e-value cut-off of 0.01), and 16 soybean *LysM-RLK* genes were obtained. Based on the above analysis results and the removed duplicates, we finally identified 27 soybean *LysM-RLK* genes (Appendix A).

### 2.2. Phylogenetic Analysis of LysM-RLK Family Proteins

To better understand the evolutionary relationship among soybean LysM-RLKs, a phylogenetic analysis of soybean and Arabidopsis LysM-RLKs were performed via the maximum likelihood method (Figure 1). Phylogenetic analysis showed that the LysM-RLK family members were divided into seven subclades (Figure 1). Since the soybean LysM-RLKs were not named, we first assigned a consistent nomenclature to the 27 LysM-RLKs based on their evolutionary relationship with Arabidopsis LysM-RLKs (Figure 1 and Appendix A).

According to the reported role of AtLYK1 and GmNFR1α/β in recognition of chitin and NFs, respectively, we speculated that the remaining three genes (*GmLYK1a*, *GmLYK1b*, and *GmLYK1c*) in subclade I might possess similar functions. Of note, only soybean LysM-RLKs were divided in subclades III, VI, and VII, suggesting that those LysM-RLK family members might have evolved novel functions in soybean.

Since Arabidopsis has only five *LysM-RLK* genes [9,10], whereas soybean has evolved 27 *LysM-RLK* genes, we attempted to decipher the co-evolutionary relationship of these *LysM-RLKs* via collinearity analysis (Figure 2). The results showed that *GmLYK4a*, *GmLYK4b*, and *GmNFR5α* in *G. max* were orthologous to *AtLYK4* in *A. thaliana*, indicating that *GmLYK4a*, *GmLYK4b*, and *GmNFR5α* may retain the same function as *AtLYK4*, which functions with *AtLYK5* in mediating chitin-triggered immune responses [14]. *GmLYK2a* in soybean is orthologous to *AtLYK2*, which encodes a LysM-RLK protein with a pseudo-kinase [15]. Synteny analysis also revealed that 17 *LysM-RLK* genes in soybean were the orthologs of 12 members in *L. japonicus*, as well as 19 *LysM-RLK* genes in soybean were the orthologs of 12 members in *M. truncatula* (Figure 2 and Appendix A). The presented data indicate that the evolutionary relationship of LysM-RLK members in legumes is comparable.

### 2.3. The Characteristics and Chromosomal Location of the LysM-RLKs

Since these LysM-RLKs belong to different subclades, we speculated that these proteins might have different properties. Indeed, the CDS lengths of 27 identified *LysM-RLKs* ranged from 1548 nucleotides to 2283 nucleotides. The lengths of LysM-RLK proteins ranged from 515 (GmNFR5β) to 760 (GmLYK6) amino acids (aa), and the molecular weight (Mw) ranged from 57.071 kDa (GmNFR5β) to 84.586 kDa (GmLYK6). These LysM-RLKs also had different predicted protein isoelectric points (PI), where the PI values of GmLYK4e, GmLYK6, GmLYK7a, and GmLYK7c were greater than 8, while the PI values of other LysM-RLK proteins ranged from 5.21 to 7.48 (Appendix A).

To demonstrate the distributions of the *LysM-RLKs* on different chromosomes, we constructed a detailed chromosome map by using the TBtools v1.120 software [30]. As shown in Figure 3, the members of the LysM-RLK family genes were distributed on fourteen chromosomes (Chr), including Chr1, Chr2, Chr7, Chr8, Chr9, Chr11, Chr12, Chr13, Chr14, Chr15, Chr17, Chr18, Chr19, and Chr20 (Figure 3). And these *LysM-RLK*s were not evenly distributed on these chromosomes, ranging from one (Chr 7, 9, 12, 17, 19, and 20) to five (Chr 2). Notably, 22 (*GmLYK3e*, *GmLYK4b*, *GmNFR5β*, *GmLYK8*, *GmLYK4a*, *GmLYK5*, *GmLYK1c*, *GmNFR1α*, *GmLYK3a*, *GmLYK7c*, *GmNFR5α*, *GmLYK4c*, *GmLYK2c*, *GmLYK3d*, *GmLYK6*, *GmNFR1β*, *GmLYK4e*, *GmLYK3c*, *GmLYK1a*, *GmLYK4d*, *GmLYK7a*, and *GmLYK7b*) of the 27 *LysM-RLKs* were localized close to the terminus region of the chromosome. In particular, the *GmLYK8*, *GmLYK3a*, *GmLYK3d*, *GmLYK6*, *GmLYK3c*, and *GmLYK4d* were closest to the end of the chromosome. In addition, the nod factor receptor gene *GmNFR1α/β* and *GmNFR5α/β* were distributed on four different chromosomes (Figure 3).

### 2.4. Conserved Protein Motifs and Domain Analysis of the LysM-RLKs

In order to understand the functions of these LysM-RLKs, we further analyzed the conserved motifs and domains (LysM and Kinase) of LysM-RLK proteins in soybean and Arabidopsis (Figure 4). As shown in Figure 4A, a total of ten conserved motifs were obtained in these LysM-RLK proteins (Figure 4A and Appendix A). Among them, motifs 2, 4, and 7 were the most conserved regions and were present in all members, indicating that these motifs may be the characteristic motifs of LysM-RLK proteins (Figure 4A). The motif analysis also revealed that members with closer relationships had a more similar arrangement of conserved motifs. For example, members of subclade I contained all motifs, while members of subclade II contained all motifs except motif 10; members of subclade V contained all motifs except motif 8; and GmLYK6, the unique member of subclade VI, contained all motifs except motif 10. All subclade III members lacked both motifs 8 and 9, and members of subclade VII lacked both motif 1 and motif 5. Finally, subclade IV members contained motifs 1, 2, 4, 7, and 10.

We then performed a conservative domain analysis to distinguish these LysM-RLKs (Figure 4B). All 27 LysM-RLK proteins possess the conserved LysM domain and Kinase domain, but the number of LysM domains varied from one to three, indicating that their functions in the reorganization of external signals may be different. According to the characteristics of these kinase domains, LysM-RLK can be divided into two types. One contains a complete kinase-conserved motif with kinase activity, while the other is missing or incomplete, resulting in pseudo-kinases [11]. We then aligned the kinase domain sequences of 27 soybean LysM-RLKs (Figure 4C), revealing that the members of subclades I, II, VI, and VII have typical kinase domains with essential conserved motifs or residues (such as G-rich, AxK, HRD, DFG, and PE), classifying them as LysM-RLK-I/LYKs with kinase activities. Conversely, all members of subclades III, IV, and V lack the G-rich motif, and some members lack AxK, HRD, DFG, and PE, classifying them as LysM-RLK-II/LYRs without kinase activities.

### 2.5. Gene Structure Analysis of LysM-RLK Genes

To gain insights into the structural evolution of the LysM-RLK family in soybean, we analyzed the gene structures of these *LysM-RLKs*. Interestingly, the exon–intron structures of different subclades varied greatly, but genes within the same subclade had similar gene structures (Figure 5A,B). The genes from subclades III, IV, and V had only one exon except for *GmLYK8* from subclade III, which contains three exons and the genes of subclade VII which also has only two (*GmLYK7b* and *GmLYK7c*) or three (*GmLYK7a*) exons. In contrast, genes in subclades I, II, and VI contained nine to twelve exons. We also noticed that one soybean unique *LysM-RLK* gene (*GmLYK7a*) has no UTR. Through analyzing and comparing the orthologous gene structures of soybean and Arabidopsis, we discovered that *AtLYK1* and *AtLYK3* consisted of 12 and 11 exons, respectively. These numbers were similar to their corresponding soybean counterparts, *GmLYK1* and *GmLYK3*, in subclades I and II. Similarly, *AtLYK4* and *AtLYK5* exhibited the presence of a single exon, just like their soybean counterparts. However, *AtLYK2* differed from the genes in subclade IV of *GmLYK2s* as it contained two exons (Appendix A). The variations observed in the gene structure of LysM-RLKs across different subclades may be associated with the transcriptional regulation during evolution.

The promoter is a crucial component of a gene, as it is where transcription is initiated and regulated. Therefore, we obtained a 2 kilobase (kb) sequence that is located upstream of the translation start codon (ATG) of all soybean *LysM-RLK* genes as a promoter for conserved motif analysis. The results showed that the motifs in the promoter region of all these LysM-RLK family genes were highly different (Figure 5C and Appendix A). Among the 10 motifs identified from 27 *LysM-RLK* genes, only motif 6 was present on all LysM-RLK promoters, suggesting that motif 6 was likely an important regulatory element for the majority of *LysM-RLKs*, which mediated the conserved biological processes. Interestingly, the motifs on the promoter of *GmLYK3c* and *GmLYK3d* were almost identical except for motif 8, which was consistent with the close evolutionary relationship of these two genes. This suggested that *GmLYK3c* and *GmLYK3d* might have similar expression patterns and participate in similar biological processes. In contrast, different motifs were identified on other *LysM-RLK* promoter sequences, indicating that these genes might have different expression patterns and divergent functions.

### 2.6. Analysis of Cis-Elements in Promoter Regions of LysM-RLK Genes

The expression of a gene is controlled by the promoter, which has various *cis*-elements regulated by transcription factors. To investigate the expression profiles and potential functions of these soybean *LysM-RLK* genes, we analyzed *cis*-acting elements in the promoters (2 kb upstream of ATG) of *LysM-RLK* genes via the PlantCARE database [31]. In addition to the TATA-box and CAAT-box, which were essential for transcriptional initiation, 23 *cis*-elements with functional prediction were identified (Figure 6A, Appendix A). These *cis*-elements could be divided into phytohormone-related (gibberellin, auxin, abscisic acid, salicylic acid, and MeJA), plant growth and development-related (meristem and endosperm expression, flavonoid biosynthetic genes regulation, circadian control, etc.), environment-related (drought-inducibility, anaerobic induction, low-temperature responsiveness, defense and stress responsiveness, etc.), and others. The findings implied that various *LysM-RLK* genes may exhibit diverse expression profiles during distinct developmental stages and in response to different biotic/abiotic stresses. In addition, we noted that genes from subclade I and subclade III, including *GmNFR1α/β* and *GmNFR5α/β*, contained more *cis*-elements in their promoter regions, indicating that the transcriptional regulation in these two subclades may be intricately complex.

Soybean can effectively establish a symbiotic relationship with rhizobia for nitrogen fixation under low nitrogen conditions. Therefore, we speculated that the expression of *GmNFR1α/β* and *GmNFR5α/β* and its homologous genes may be regulated via nitrogen. To validate this hypothesis, we examined the *cis*-elements associated with plant response to nitrogen (NIN/NLP binding site, NBS) and symbiotic nodulation (nodule specificity, AAAGAT/CTCTT) [32,33,34]. The results showed that all *LysM-RLK* gene promoters contained nodule specificity elements, while 51.86% of the *LysM-RLK* gene promoters (*GmLYK4c*, *GmLYK4b*, *GmLYK4a*, *GmLYK5*, *GmLYK4d*, *GmLYK2a*, *GmLYK2c*, *GmLYK8*, *GmNFR5α*, *GmNFR1β*, *GmLYK3f*, *GmLYK3e*, *GmLYK3a*, and *GmLYK3b*) had NBS (Figure 6B), suggesting these genes potentially played roles in both symbiotic nitrogen fixation and nitrogen response processes.

### 2.7. Expression of LysM-RLK Family Genes in Response to Rhizobia

To further explore the potential functions of these *LysM-RLKs*, we retrieved the expression data of 21 *LysM-RLK* genes (the remaining six members were not found in the database) via the eFP website and analyzed their expression profiles in response to rhizobia. The results demonstrated that the gene expression levels were particularly low in subclades II, IV VI, and VII, whereas a majority of the individuals in subclades I, III, and V had a substantially high expression (Figure 7). In particular, the expression of *GmNFR5α* had increased at 12 h after inoculation (HAI). Beside *GmNFR5α, GmLYK4d* was also up-regulated at 12 HAI, which indicates *GmLYK4d* may also play roles during the symbiosis establishment. The expression of *GmNFR1α* was suppressed by rhizobia at 12, 24, or 48 HAI, while the expression level of *GmNFR1β* remained very low, which was consistent with the previous findings [23]. We also found that the expression of several subclade V members (*GmLYK4c*, *GmLYK4b, GmLYK5*, and *GmLYK4e*) was slightly inhibited by rhizobia, while the expression of *GmLYK4a* was slightly upregulated at 12 HAI, and then reduced along with inoculation. Although *GmLYK8* and *GmLYK1b* were expressed at high levels in the root hairs, their expression levels are basically unregulated by rhizobia.

### 2.8. Expression of LysM-RLK Family Genes under Different Nitrogen Treatments

The soybean–rhizobia symbiosis was established under low nitrogen conditions, so we specifically focused on the expression pattern of the *LysM-RLK* genes under different nitrogen treatments based on the RNA-seq data from the JGI Plant Gene Atlas (Figure 8) [35]. The results showed that the expression of most of the members from subclade V (*GmLYK4c*, *GmLYK4a*, *GmLYK4d*, and *GmLYK4e*) and *GmLYK1b* from subclade I were induced via various nitrogen in roots, especially *GmLYK4a* and *GmLYK4d.* On the other hand, the expression of *GmLYK4b* was induced via ammonium and nitrate nitrogen but was inhibited by urea. In contrast, the expressions of *GmLYK5, GmLYK3a*, *GmLYK3b* were inhibited by various nitrogen in roots, and *GmLYK7c* was inhibited by nitrate or urea treatment (Figure 8A). The expression patterns of *LysM-RLKs* in leaves and roots were significantly different, with only *GmLYK5* and *GmLYK7c* showing a significant inhibition of expression in response to nitrogen treatment, while the rest of the genes did not exhibit any significant response to nitrogen treatment (Figure 8B).

Combined with the expression profiles under rhizobia treatment, we found that the genes from subclade V (*GmLYK4c*, *GmLYK4b*, *GmLYK4a*, *GmLYK5*, *GmLYK4d*, and *GmLYK4e*) in roots were regulated by both rhizobia and nitrogen treatment, suggesting that genes from subclade V may function in coordinating the nodulation process and plant nitrogen response.

### 2.9. Verification of LysM-RLKs Expression in Response to Rhizobia and Nitrate

To validate whether *LysM-RLKs* are responsive to rhizobia and nitrogen treatments, we analyzed the expression of five genes from subclade V (*GmLYK4c*, *GmLYK4b*, *GmLYK4a*, *GmLYK5*, *GmLYK4d*, and *GmLYK4e*) under rhizobia and nitrate treatments in roots by performing quantitative PCR (Figure 9 and Appendix A). The results showed that the expression levels of *GmLYK4a*, *GmLYK4c*, and *GmLYK5* in roots were inhibited by rhizobia at 3 days post-inoculation (DPI) (Figure 9A,C,F), and the expression of *GmLYK4d* in roots was induced by rhizobia at 1 DPI (Figure 9D), which is basically consistent with the previous analysis results. *GmLYK4e* was induced in roots by rhizobia at 1 DPI and then descended to a lower level at 3 DPI (Figure 9E), while the expression of *GmLYK4b* in roots was not affected by rhizobia (Figure 9B).

We also discovered that these selected genes were responsive to nitrate treatments (Figure 9G–L). In roots, the expressions of *GmLYK4a*, *GmLYK4b*, *GmLYK4c*, and *GmLYK4e* were significantly induced via various concentrations of nitrate treatment at 1 day after treatments (Figure 9G–I,K). However, these genes continuously maintain a high expression level in roots via both moderate (5 mM) and high (10 mM) nitrate at 3 days after treatments (Figure 9G–I,K). *GmLYK4d* was upregulated at 1 day after high nitrate treatments, and its expression was markedly induced via various concentrations of nitrate at 3 days after treatment (Figure 9J). In comparison, *GmLYK5* was repressed by low (1 mM) and moderate levels of nitrate at 1 day after treatment and continuously decreased with prolonged low-nitrogen treatments but was induced via moderate and high levels of nitrate at 3 days after treatment (Figure 9L).

These results confirm that the expression of subclade V genes are responsive to both rhizobia and nitrate, indicating that *GmLYK4a*, *GmLYK4b*, *GmLYK4c*, *GmLYK4d*, *GmLYK4e*, and *GmLYK5* play roles in both symbiotic nitrogen fixation and nitrogen response processes.

## 3. Discussion

LysM-RLK is found widely in plants and plays an important role in plant defense against pathogens and the establishment of symbiosis relationships between plants and microorganisms [36]. The functional diversity of LysM-RLK is primarily determined via its extracellular LysM domain and its intracellular kinase domain [6,37]. In this study, 27 *LysM-RLK* genes were identified in soybean and assigned consistent nomenclatures based on their evolutionary relationship with Arabidopsis *LysM-RLKs*. A comprehensive analysis of LysM-RLK family genes in soybean was conducted, encompassing classification, chromosomal distribution, phylogenetic tree, and expression profiles concerning rhizobia and nitrogen treatment. The outcomes of our study provide detailed insights into soybean LysM-RLKs and highlight the role of subclade V members in both symbiotic nitrogen fixation and nitrogen response.

Despite being diploid, *G. max*, possessing an octoploid-like genome, is expected to possess a greater number of *LysM-RLKs* compared to the diploid species. In our study, we utilized rigorous criteria and the latest soybean genome information to identify 27 LysM-RLKs in soybean, whereas *A. thaliana* and *L. japonicus* had 5 and 20 LysM-RLKs, respectively [10,38]. Phylogenetic analysis of *G. max* and *A. thaliana* LysM-RLKs revealed that only soybean LysM-RLKs were classified in subclades III, VI, and VII, implying their potential evolution of novel functions in soybean. Notably, the soybean Nod factor receptors GmNFR1α/β and Arabidopsis AtLYK1 share a subclade and exhibit similar protein structures, implying that they have the same functionality. The specificity of LysM-RLK binding to extracellular glycans is mainly determined via the N-terminal LysM1 domain [22]. Indeed, GmNFR1α/β and AtLYK1 prefer to recognize NFs and chitin, respectively, reflecting the diversity and difference of the LysM motif between GmNFR1α/β and AtLYK1 [22]. Conversely, Arabidopsis employs a mechanism to perceive the rhizobia NFs, which effectively inhibits the immune response triggered by MAMPs, leading to a decreased abundance of PRRs on the plasma membrane that are responsible for MAMPs’ recognition [39]. The subclade of the GmNFR5α/β is unique to soybean, although *GmNFR5α* and *AtLYK4* were collinear gene pairs. Both GmNFR5α and AtLYK4 proteins lack kinase activity and function as co-receptors with GmNFR1α and AtLYK1, respectively [14,23,40].

In total, 27 soybean *LysM-RLK* genes were classified into seven subclades. The number of LysM-RLKs ranged from one (subclade VI) to six (subclades II and V), and different subclades exhibited distinct gene structures. Interestingly, genes in subclades III, IV, and V typically consisted of a single exon, except for *GmLYK8* in subclade III which contained three exons. These genes encoded LysM-RLK with pseudo-kinase lacking in the G-rich motif. Therefore, based on the classification principle of Gust et al. [2], the members of subclades III, VI, and V belong to LysM-RLK-II. Moreover, genes in the subclade VII also have either two (*GmLYK7b* and *GmLYK7c*) or three (*GmLYK7a*) exons, yet all GmLYK7 proteins with a canonical kinase domain belong to the LysM-RLK-I category. In contrast, members of subclades I, II, and VI comprised twelve genes containing nine to twelve exons, and the encoded LysM-RLK featured the classical kinase domain, categorizing them as LysM-RLK-I [2]. Lohmann et al. further classified LysM-RLK into LYS-I, LYS-II, and LYS-III in 2010, considering the evolutionary distance, kinase characteristics, and gene structures [38]. According to this classification, members of subclades I and II were assigned to LYS-1, members of subclades III, VI, and V were assigned to LYS-II, members of subclade VII were assigned to LYS-III, and GmLYK6 did not fit into any of these groups.

The functional disparities of a gene are determined via the precise gene expression site and the magnitude of its promoter’s activity. Indeed, we observed substantial diversity in the promoter sequences of these *LysM-RLKs*. Almost all *LysM-RLK* promoters contained *cis*-elements associated with plant development and the responses to environmental factors, including phytohormones (e.g., MeJA, abscisic acid, gibberellin, etc.) and abiotic stresses (e.g., low temperature, drought, etc.). Notably, the number of *cis*-elements related to MeJA, abscisic acid, and gibberellin was significantly higher compared to others, as indicated in Figure 4. Jasmonic acid and abscisic acid are well known for their roles in plant responses to biotic and abiotic stresses, respectively [41,42]. The abundance of these *cis*-elements in the promoter regions of soybean *LysM-RLKs* suggests their crucial role in facilitating the plant’s adaptation to environmental fluctuations. Moreover, previous studies have explored the responses of LysM genes to biotic and abiotic stresses in wheat and banana [43,44]. However, the functional implications of the candidate *LysM-RLK* genes, identified via our bioinformatic analysis, in biological and abiotic stress remain uncertain, warranting further investigation and deliberation.

Soybean can effectively establish a symbiotic relationship with rhizobia for nitrogen fixation in conditions of low nitrogen. Our findings indicated that all *LysM-RLK* gene promoters contain nodule-specificity elements, while 51.86% of the *LysM-RLK* gene promoters (*GmLYK4c*, *GmLYK4b*, *GmLYK4a*, *GmLYK5*, *GmLYK4d*, *GmLYK2a*, *GmLYK2c*, *GmLYK8*, *GmNFR5α*, *GmNFR1β*, *GmLYK3f*, *GmLYK3e*, *GmLYK3a*, and *GmLYK3b*) had NBS (Figure 6B), suggesting their involvement in both symbiotic nitrogen fixation and nitrogen response processes. By analyzing the RNA-seq data of these genes, it revealed that the genes from subclade V (*GmLYK4c*, *GmLYK4b*, *GmLYK4a*, *GmLYK5*, *GmLYK4d*, and *GmLYK4e*) were regulated by both rhizobia and nitrogen treatment in roots (Figure 7 and Figure 8), suggesting their role in coordinating the nodulation process and plant nitrogen response. RT-qPCR analysis demonstrated that the expression of *GmLYK4a* and *GmLYK4c* in roots was inhibited by rhizobia but induced via varying concentrations of nitrate (Figure 9), indicating that *GmLYK4a* and *GmLYK4c* function as regulators in nitrate response and symbiotic nitrogen fixation. The expression of *GmLYK4b* in roots was induced via nitrate, but not regulated by rhizobia (Figure 9), suggesting its involvement in nitrogen response. The expression of *GmLYK4d* was induced by rhizobia at the early stage of inoculation, while its induction by nitrate requires longer treatment time to manifest (Figure 9), signifying its role in different stages of rhizobia response and nitrogen response. Both rhizobia and nitrate treatment can induce the expression of *GmLYK4e* in roots within a short duration (Figure 9), suggesting its potential involvement in both rhizobia and nitrate sensing. In the early stage of nitrate treatment, the expression of *GmLYK5* in roots was inhibited by low nitrate and moderate nitrogen, but with the extension of treatment, its expression tended to be induced via high nitrate (Figure 9), indicating its sensitivity to different nitrate concentrations. The expression of *GmLYK5* in roots was also inhibited by rhizobia (Figure 9), indicating that *GmLYK5* has diverse functions in different stages of rhizobia and nitrogen response. These results collectively indicate that *GmLYK4a*, *GmLYK4c*, *GmLYK4d*, *GmLYK4e*, and *GmLYK5* function in coordinating the nodulation process and plant nitrogen response.

To summarize, a comprehensive analysis was conducted on the genes and proteins of the LysM-RLK family in soybean. The outcomes of our study establish a fundamental basis for exploring the function of LysM-RLK family genes in soybean development and environmental adaptation. Furthermore, our analysis unveiled the previously undocumented role of specific LysM-RLKs in coupling both the symbiotic nitrogen fixation process and responding to nitrogen stimuli. Moving forward, the identification of the biological functions associated with LysM-RLK family genes holds promise in generating novel insights for optimizing soybean genetics and breeding strategies.

## 4. Materials and Methods

### 4.1. Sequence Retrieval and Filtering

Soybean (*Glycine max* Wm82.a4.v1) genome, GFF3 file, and protein sequences were downloaded from the Phytozome database (https://phytozome-next.jgi.doe.gov/ (accessed on 5 April 2023)). The hidden Markov model (HMM) file of the LysM domain (PF01476) and the Kinase domain (PF00069) were downloaded from the Pfam database (http://pfam.sanger.ac.uk/ (accessed on 10 April 2023)) [45]. Simple HMM search of the TBtools software was used to obtain LysM-containing proteins and Kinase-containing proteins [30]. The known A. thaliana LysM-RLK protein sequences were used as queries for a Blast search in the soybean database, and all retrieved sequences that showed low reliable similarities, with an identity cut-off of 40% and an e-value cut-off of 0.001, were excluded from the analysis. The filtered proteins were further confirmed via the NCBI CD-Search tool (e-value cut-off of 0.01) (https://www.ncbi.nlm.nih.gov/Structure/cdd/wrpsb.cgi (accessed on 20 April 2023)) [46].

### 4.2. Phylogenetic Analysis 

The amino acid sequences of selected LysM-RLKs were aligned via the MEGA-X software and the phylogenetic tree was constructed via the maximum likelihood method (ML) or the neighbor-joining method (NJ). Bootstrap analysis was calculated for 1000 replicates [47]. The evolutionary tree was visualized using the web-based tool iTOL (https://itol.embl.de/ (accessed on 20 April 2023)) [48].

### 4.3. Synteny Analysis of the LysM-RLK Genes

A collinearity analysis of the *LysM-RLK* genes was performed using TBtools software. The genomes of *A. thaliana*, *L. japonicus*, *M. truncatula*, and *G. max* were compared using One Step MCScanX-Super Fast with default parameters (e-value < 1 × 10^−3^). TBtools was used to map the collinearity graphs and highlight the *LysM-RLK* genes on the graphs [30].

### 4.4. The Chromosomal Location of the LysM-RLK Genes

TBtools software was utilized to display the gene density and chromosomal locations of soybean *LysM-RLKs* based on the GFF3 file and gene ID [30].

### 4.5. Conserved Motif, Protein Conserved Domain, and Gene Structure Analysis

The promoter sequences, CDSs, genomic sequences, and amino acid sequences of *LysM-RLKs* were obtained from the Phytozome database. The conserved motifs in LysM-RLK amino acid sequences and the promoter sequences of *LysM-RLKs* were analyzed via the MEME (https://meme-suite.org/meme/tools/meme (accessed on 15 April 2023)) online tool [49]. The conserved domains of LysM-RLK proteins were analyzed via the NCBI CD-search tool (https://www.ncbi.nlm.nih.gov/Structure/cdd/wrpsb.cgi (accessed on 20 April 2023)) [46]. The exon–intron structures of *LysM-RLKs* were visualized using the TBtools software based on the soybean GFF3 file, and TBtools software was also used to integrate the conserved motifs, protein conversed domains, and gene structures [30].

### 4.6. Cis-Element Analysis of LysM-RLK Promoter Regions

The promoters (2 kb upstream of ATG) of *LysM-RLKs* were obtained from the Phytozome database. The *cis*-elements in the promoter regions of 27 *LysM-RLKs* were analyzed using the PlantCARE online website (https://bioinformatics.psb.ugent.be/webtools/plantcare/html/ (accessed on accessed on 5 May 2023)). The data was visualized via TBtools [30].

### 4.7. Expression Analysis of the LysM-RLKs in Response to Rhizobia and Nitrogen

The expression data of *LysM-RLKs* in response to rhizobia were downloaded from the eFP (Soybean eFP Browser (utoronto.ca)) website, and the *LysM-RLKs’* gene expression data in response to different nitrogen treatments were obtained from the JGI Plant Gene Atlas (https://plantgeneatlas.jgi.doe.gov/ (accessed on 5 May 2023)) [35,50]. Gene expression heatmaps were constructed via TBtools software [30].

### 4.8. Plant Materials and Growth Conditions

Seeds from soybean cultivar Williams 82 (Wm82) were sown in vermiculite soaked in a nitrogen-free BD nutrient solution, as described by Broughton and Dilworth, and cultured in a greenhouse under controlled conditions with a 16 h light/8 h dark at 26 °C, a light intensity of 200 μmol/m^2^/s, and 60% relative humidity) [51]. To analyze the *LysM-RLKs’* expression pattern under rhizobia inoculation, 7-day-old plants were inoculated with *Bradyrhizobium diazoefficiens* USDA110 (OD_600_ = 0.08, 30 mL per plant) and roots were collected at 0 DPI, 1 DPI, and 3 DPI. For samples treated with various concentrations of nitrate, 7-day-old soybean plants were treated with 0 mM, 1 mM, 5 mM, and 10 mM of nitrate nitrogen, respectively. The roots were taken at 1 day and 3 days after nitrate treatment for subsequent RT-qPCR experiments.

### 4.9. RNA Extraction and RT-qPCR

To verify the expression of *LysM-RLKs*, total RNA was extracted from different samples using a TRIpure reagent (Aidlab Biotechnologies, Beijing, China). cDNA was synthesized from the RNA via reverse transcription reagent kit (YEASEN (Shanghai, China)) and the Hifair^®^ II 1st Strand cDNA Synthesis SuperMix for qPCR (gDNA digester plus). Real-time quantitative PCR was performed using the Hieff qPCR SYBR Green Master Mix Kit (No Rox) (YEASEN). *GmELF1b* was used as an internal gene [52].

## Figures and Tables

**Figure 1 ijms-24-13621-f001:**
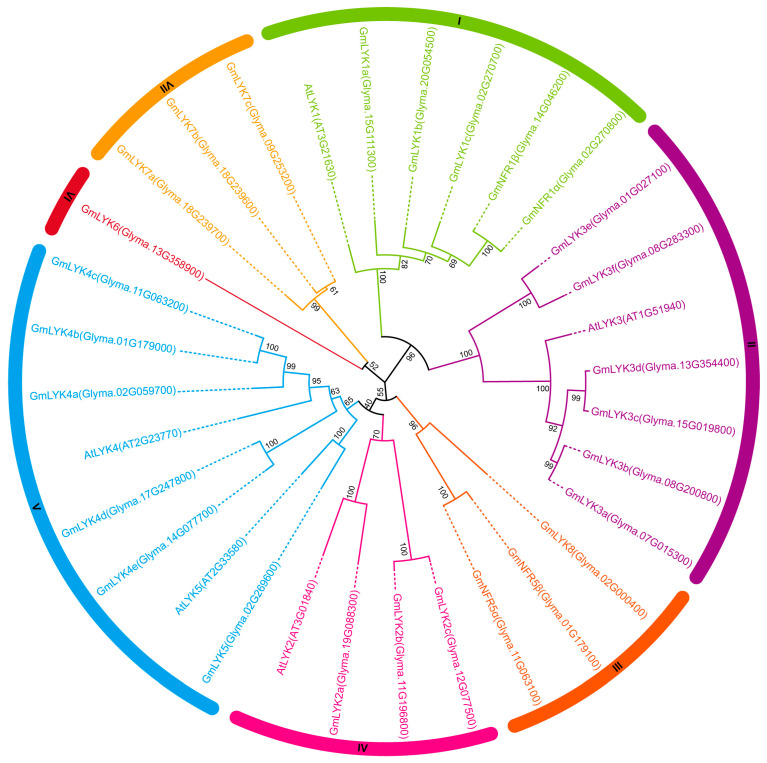
Phylogenetic tree of LysM-RLK proteins in *G. max* and *A. thaliana* were constructed via the maximum likelihood method. The tree was constructed with 1000 bootstrap replications. Proteins from different subclades (I–VII) were designated with different color zones.

**Figure 2 ijms-24-13621-f002:**
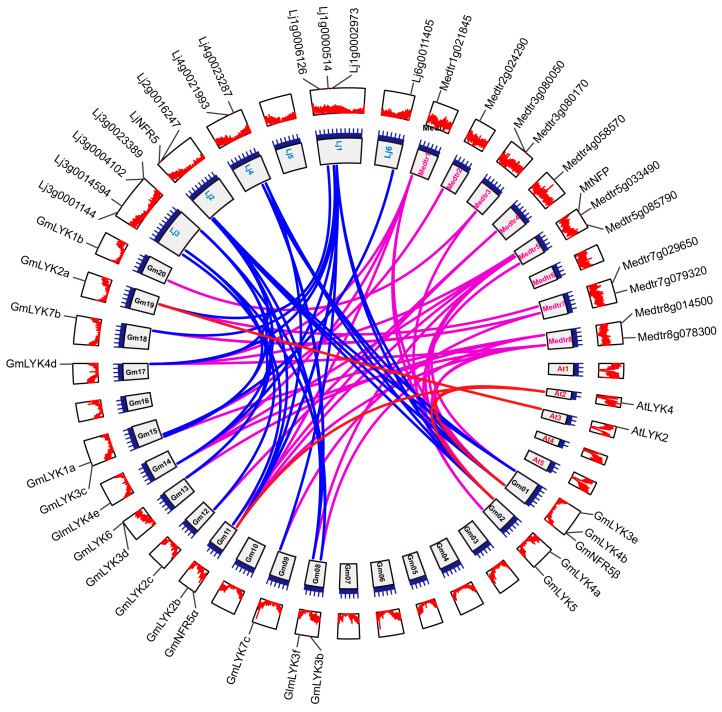
Synteny analysis of *LysM-RLK* genes among *G. max*, *A. thaliana*, *L. japonicus*, and *M. truncatula*. The genomics sequences were aligned using One Step MCScanX-Super Fast of TBtools. The circos tracks display chromosome length, gene location, and gene density, arranged from inner to outer. The red lines indicate the syntenic *LysM-RLK* gene pairs between soybean and Arabidopsis, blue lines indicate the syntenic pairs between soybean and Lotus, and purple lines depict syntenic pairs between soybean and Medicago.

**Figure 3 ijms-24-13621-f003:**
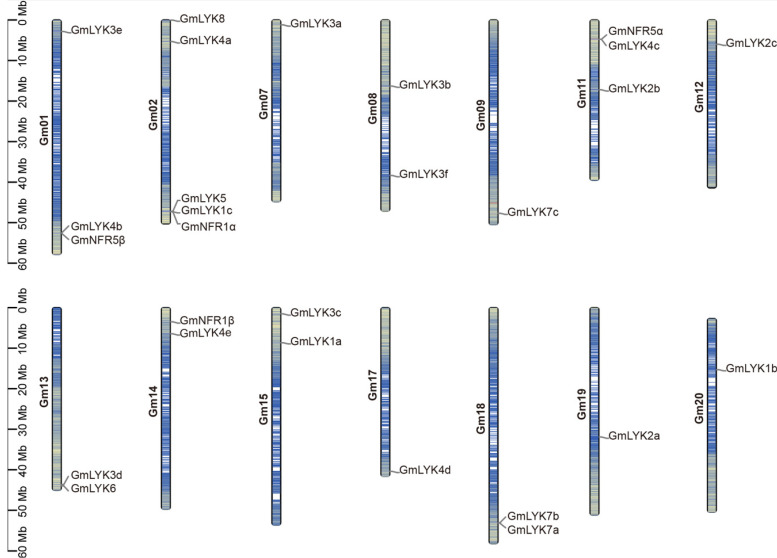
Chromosomal distribution analysis of *LysM-RLK* genes.

**Figure 4 ijms-24-13621-f004:**
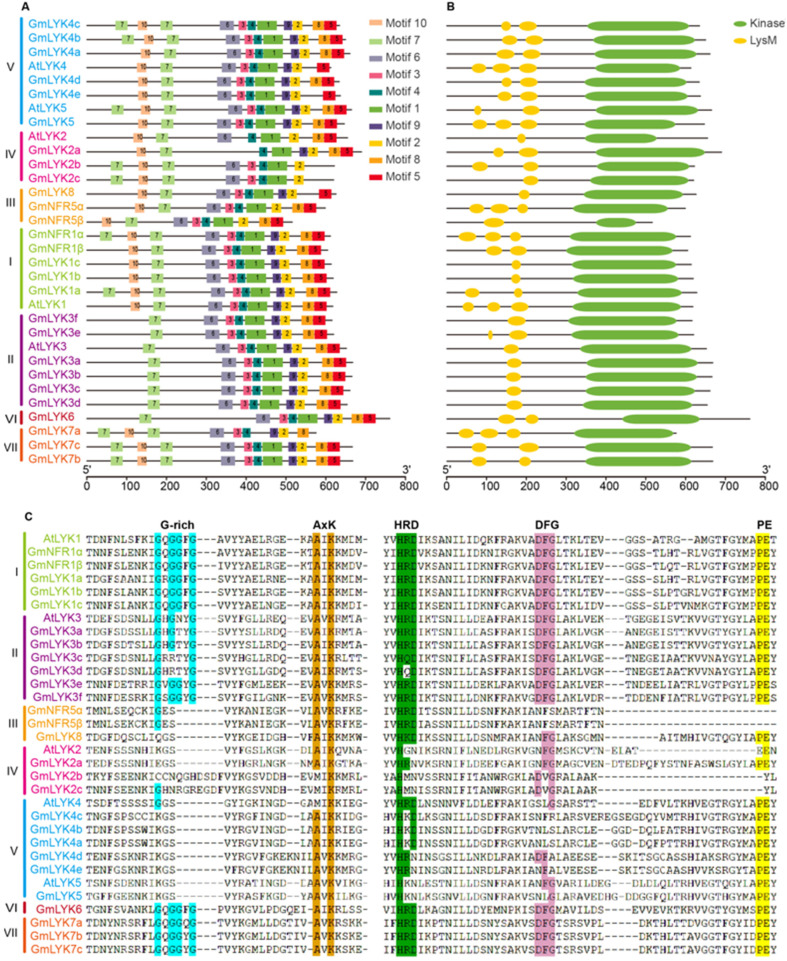
Analysis of conserved protein motifs and domains in LysM-RLKs from soybean and Arabidopsis. (**A**) Conserved motif analysis of LysM-RLK amino acid sequences via the MEME online website. Various motifs were represented by distinct numbers. (**B**) Conserved domain analysis of LysM-RLK proteins via the NCBI CD-search tool. (**C**) Amino acid alignment of the kinase sequences of LysM-RLKs. Conserved motifs or residues (G-rich, AxK, HRD, DFG, and PE) that are important for kinase activity are highlighted in different colors.

**Figure 5 ijms-24-13621-f005:**
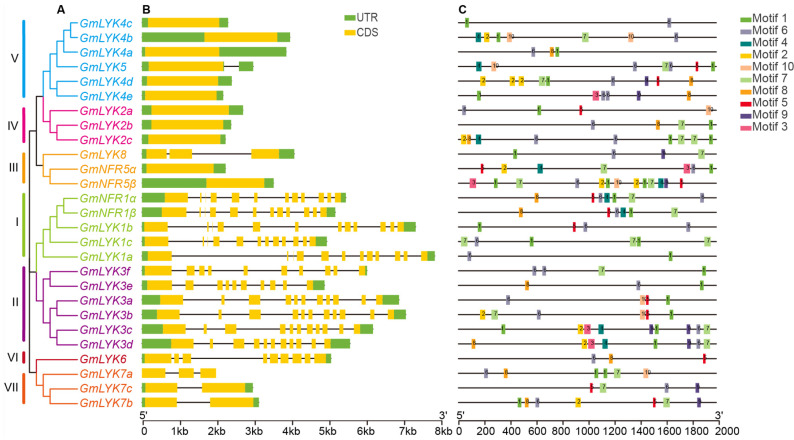
Gene structures and conserved promoter motif analysis of the *LysM-RLKs* in soybean. (**A**) Phylogenetic tree analysis of LysM-RLKs in soybean. Neighbor joining trees were constructed via MEGA-X-10.0.2 software with 1000-fold bootstrap resampling. The branches in different colors signify distinct subclades. (**B**) Gene structure of *LysM-RLK* genes was determined via the utilization of the GSDS 2.0 online website. Exon regions were represented by yellow boxes, UTR regions by green boxes, and intron regions by black lines. (**C**) The conserved motifs of *LysM-RLK* promoters in soybean were analyzed via the MEME website. Each conserved motif was assigned a specific number. The 2 kb promoters of *LysM-RLK* genes in soybean were obtained from the Phytozome database.

**Figure 6 ijms-24-13621-f006:**
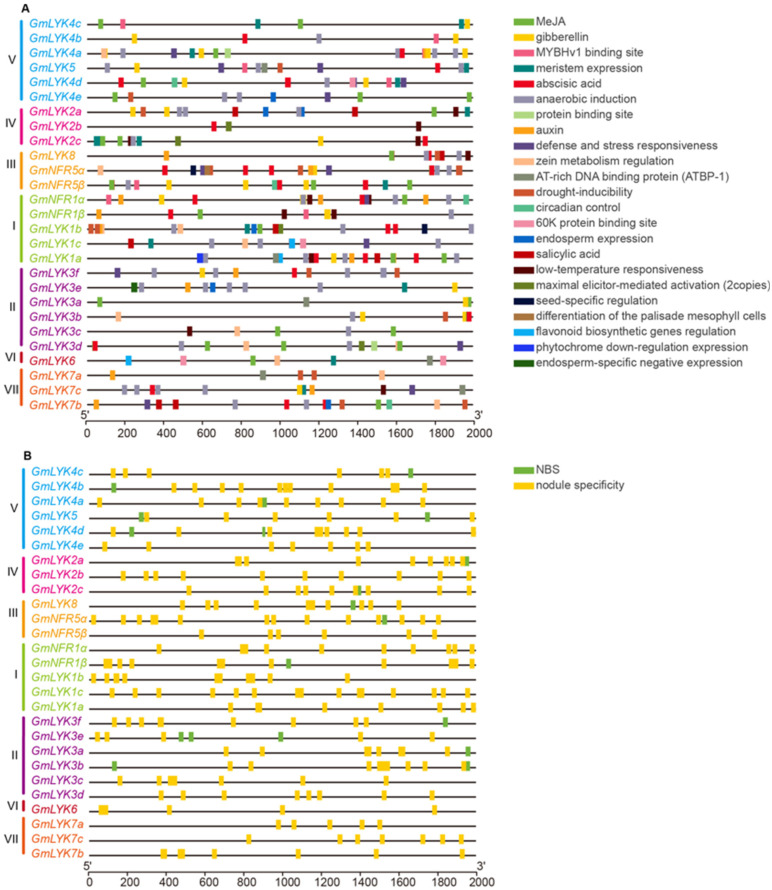
Prediction of *cis*-element in promoter regions of *LysM-RLKs*. 2 kb upstream sequences (from the translation initiation site, ATG) were downloaded from Phytozome and used to analyze the *cis*-elements. (**A**) The *cis*-acting elements of 27 *LysM-RLK* promoters in soybean were analyzed via the PlantCARE website. (**B**) *Cis*-acting elements of NBS sites and nodule specificity in *LysM-RLK* promoters were analyzed. *Cis*-acting elements with similar regulatory functions were indicated by the same color.

**Figure 7 ijms-24-13621-f007:**
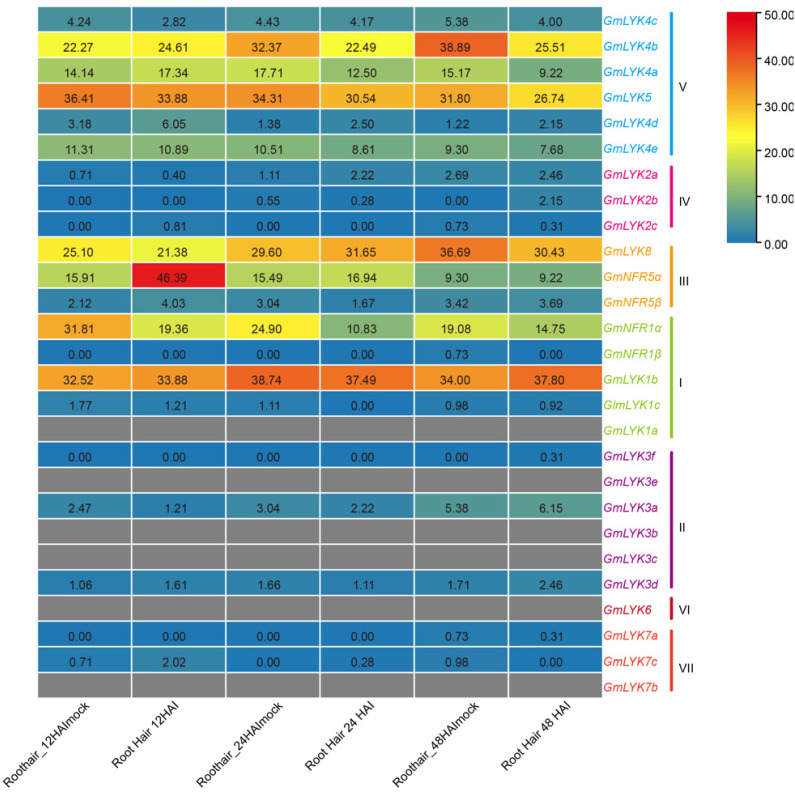
A heatmap depicting the expression levels of *LysM-RLK* genes in response to rhizobia, based on the RNA-seq data. The RNA-seq data were downloaded from the eFP website. The gradient of colors ranging from blue to red corresponds to varying levels of gene expression, with blue indicating low expression and red indicating high expression. HAI represents hours after inoculation and HAImock stands for the absence of rhizobia inoculation. The numerical values denote the expression level, and the gray box indicates the unavailability of expression data.

**Figure 8 ijms-24-13621-f008:**
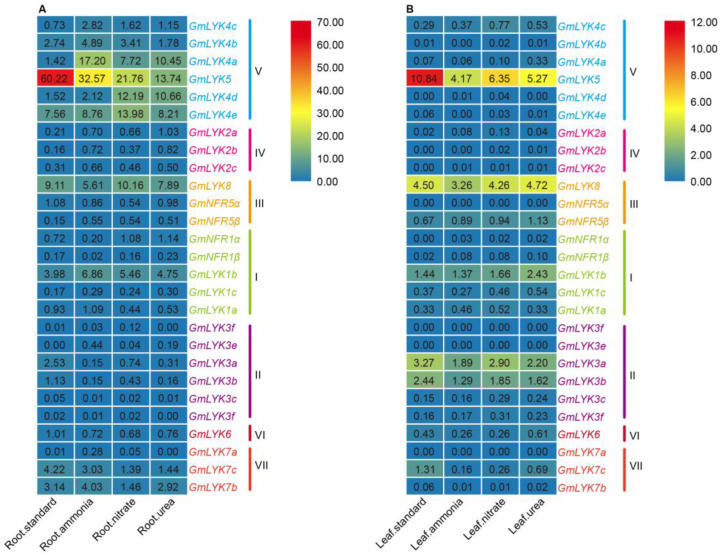
Heatmap of *LysM-RLK* gene expression in response to various forms of nitrogen treatment in both roots (A) and leaves (B). The *LysM-RLK* expression data were downloaded from the JGI Plant Gene Atlas, and the color scale ranging from blue to red indicating low to high expression levels. The numerical values indicate the level of expression.

**Figure 9 ijms-24-13621-f009:**
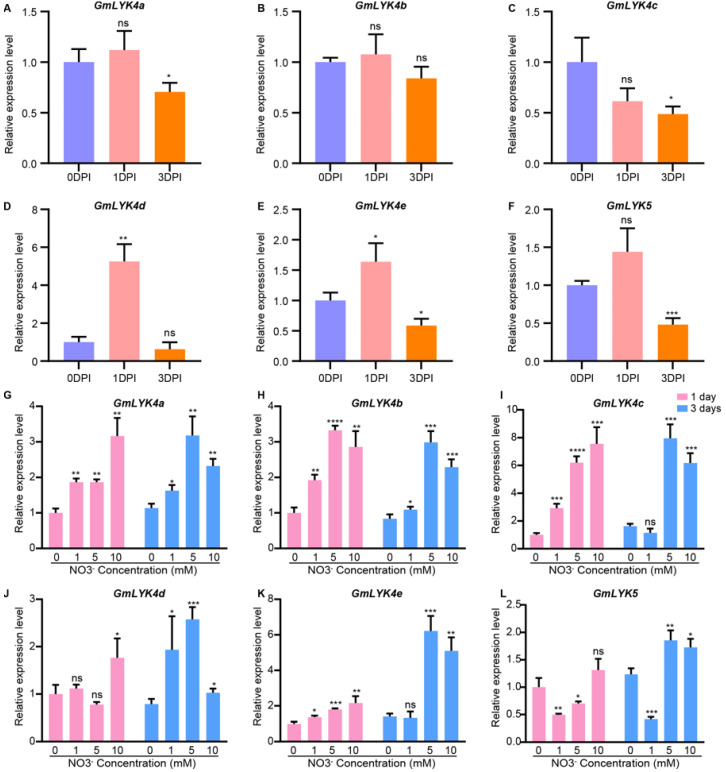
Gene expression analysis of *LysM-RLKs* in response to rhizobia and nitrate treatments. (**A**–**F**) The expression of *LysM-RLKs* in roots was assessed using RT-qPCR at 0 DPI and 3 DPI to measure the response to rhizobia. (**G**–**L**) The expression levels of *LysM-RLKs* in roots were analyzed under different concentrations of nitrate (0 mM, 1 mM, 5 mM, and 10 mM) at 1 day (purple bars) and 3 days (blue bars). *GmELF1b* was used as a reference gene, and the relative expression levels of *LysM-RLKs* were calculated using the 2^−ΔΔCq^ method. The data were analyzed using student’s *t*-test to determine significant differences (*, *p* < 0.05; **, *p* < 0.01; ***, *p* < 0.001; ****, *p* < 0.0001; ns, no significant differences).

## Data Availability

The datasets used and/or analyzed during the current study are available from the corresponding author on reasonable request. However, most of the data is shown in Appendix A.

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
