# Peer review of "Genome-Wide Identification of the Soybean LysM-RLK Family Genes and Its Nitrogen Response"

_ijms, 2023, doi:10.3390/ijms241713621_

Round 1
Reviewer 1 Report
The present research provides new insights into the mechanisms underlie the symbiotic relation rhizobia and the soybean in nitrogen fixation and nitrogen response, revealing the role of LysM-RLK family members in the soybean genome. The manuscript is clearly written and the bibliography consulted is adequate.
Minor changes:
Line 183 - replace semicolon with full stop (except motif 10; All subclade)
Line 387 - erase the comma (Gust et al.,[2] ).
Line 410 - add a period (banana[43,44] However, the).
Line 421 – change Figure to Figures.
Reviewer 2 Report
The authors present a comprehensive search for LysM-RLK family genes in the soybean genome, and follow through with an experimental analysis via RT-PCR of the expression response of a soybean-specific clade of these genes to rhizobial innoculation and nitrate. They find that several of the genes investigated are upregulated by nitrogen and are also inhibited by rhizohbia. This is an interesting study that makes good contributions. I found it surprising that such an important family of genes had not already been collectively characterized in soybean more than a decade after that organism's genome had been sequenced.
A few minor comments and suggestions follow:
- In Figure 1, it would be really helpful to print the bootstrap values at the branch points in the tree so we can see the confidence in each clade.
- Lines 130-131 “collinearity analysis (Figure 2). The results showed that GmLYK4a, GmLYK4b and GmNFR5α in G. max were orthologous to AtLYK4 in A. thaliana”. GmLYK4a and GMNFR5α are clearly marked in Figure 2, but GmLYK4b does not appear to be labeled in the figure.
- Lines 214-216 “The variations in the gene structure of LysM-RLKs across different subclades may be associated with their regulation at both transcriptional and post-transcriptional levels.” It also could simply be a result of distinct evolutionary histories. It would be very instructive for you to show a comparison to the gene structure of the orthologous genes in Arabidopsis. For example, do AtLYK4 and AtLYK5 have only one exon like their counterparts in soybean, and do AtLYK1 and AtLYK3 have many exons like their syntenic soybean partners?
- Lines 364-365 “G. max, being an ancient tetraploid, is expected to possess a greater number of LysM-RLKs compared to diploid species.” That is far from explaining the difference of 5 genes in Arabidopsis vs. 27 in soybean. In actuality, soybean is an ancient octaploid. An ancient polyploid event occurred in a legume progenitor to soybean, lotus, Medicago and peanut 55 mya (see, e.g., https://doi.org/10.1093%2Fmolbev%2Fmsu296). After that, there was a glycine-specific whole-genome duplication 5-13 Mya (discussed in same reference and many others). That would bring us up to 5 x 2 x 2 = 20 for a baseline for soybean. Add soybean- and lotus-specific expansions of the families and we could arrive at the large numbers of genes observed today.
